# Effects of aerobic exercise on body self-esteem among Chinese college students: A meta-analysis

**Junwen Shu, Tianci Lu, Baole Tao, Hanwen Chen, Haoran Sui, Lingzhi Wang, Ye Zhang, Jun Yan**  *

College of Physical Education, Yangzhou University, Yangzhou, Jiangsu, China

* yanjun@yzu.edu.cn

**Data Availability Statement:** All relevant data are within the paper and its Supporting Information files.

**Funding:** The author(s) received no specific funding for this work.

## Abstract

### Objective

To investigate the effect of aerobic exercise on five dimensions of physical self-worth, exercise capacity, physical condition, physical attractiveness, and physical quality in body self-esteem of Chinese college students.

### Methods

By searching PubMed, Web of Science, Cochrane Library, CNIK database, VIP database, WANFANG database platform, we searched for the subject terms or keywords "body self-esteem", "Chinese college students", "Systematic evaluation", "Aerobic exercise", "Exercise intervention", "Meta-Analysis". The search method was a combination of subject terms and keywords and title, and the search period was from database creation to The search was conducted from database creation to May 2022. A total of 3221 articles were searched, and 9 articles were included in the study through repeated screening. Risk of bias was assessed with Cochrane and the quality of studies in the literature was assessed using Grade pro software. The outcome indicators of the included literature were analysed using review manager 5.4 software and StataMP 17.0 software.

### Results

Nine papers including 1613 subjects were included. results of Meta-analysis showed that aerobic exercise was effective in improving physical self-worth ($WMD = 1.46$, 95% $CI$: 1.08–1.83, $p<0.001$), improving exercise capacity ($WMD = 1.62$, 95% $CI$: 1.23–2.01, $p<0.001$), improving physical attractiveness ($WMD = 1.32$, 95% $CI$: 0.98–1.67, $p<0.001$), improved physical condition ($WMD = 1.32$, 95% $CI$: 0.98–1.67, $P<0.001$), improved physical fitness ($WMD = 1.51$, 95% $CI$: 1.07–1.95, $P<0.001$). The differences were all statistically significant.

### Conclusion

Aerobic exercise can effectively improve the body self-esteem of Chinese college students. In exercise, male students pursue is athletic ability and physical fitness, and female students

**Competing interests:** The authors have declared that no competing interests exist.

pursue is the sense of physical self-worth and physical attractiveness. Aerobic exercise has a greater increase in body self-esteem for obese or Obese college students. Aerobics and physical dance are the most cost-effective for improving body self-esteem. Medium-intensity relative to low-intensity exercise was effective for body self-esteem intervention. A single exercise session of 90 minutes was more effective than a single 30-minute session in boosting body self-esteem, and the overall intervention duration of 16 weeks was more effective than 10 weeks.

## 1. Introduction

With the increasing number of college students graduating today, the rapid social and economic development, the accelerated pace of life, and the normalization of the global new crown epidemic, the employment situation is becoming increasingly challenging, and for many Chinese college students, going to college is a stressful time [1]. In addition to dealing with academic pressure, they also have to deal with their own relationships with teachers, classmates, roommates and lovers, and some college students also have to deal with a series of pressures such as separation from their families of origin and competition for employment, which can have a negative impact on mental health [1–3]. Therefore, solving the mental health problems of Chinese university students and promoting the overall development of their physical and mental health have become the top priorities of Chinese universities [4].

Self-esteem is considered by psychologists to be the variable that best predicts changes in mood and personality [5]. Body self-esteem refers to an individual's emotional feelings about different aspects of their body, either positive or negative, based on their own evaluations. It is a specific area under overall self-esteem and is an important component of self-esteem and mental health [6]. Among them, body self-esteem includes five dimensions: physical self-worth, athletic ability, physical condition, physical attractiveness, and physical qualities [7]. Body self-worth is an important part of the self. Body self-worth is closely linked to the whole self, plays an integrating role in the whole self, and refers to the degree to which individuals are satisfied with various aspects of their bodies such as their looks, athletic ability, body size [8] (Chen et al.,2004). Athletic ability refers to the ability of people to participate in sports and training, is a comprehensive performance of human body form, quality, function, skills and mental ability and other factors, from the biochemical point of view of analysis, athletic ability depends mainly on the supply, transfer and utilization of energy in the process of exercise [9] (Patel et al.,2021). There are four types of physical conditions, namely: healthy state, sub-healthy state, precursor state before the onset of disease, and disease onset state [10] (Rivera et al.,2019). Physical attractiveness is defined as a dimension of psychological evaluation of the state and degree of beauty of the human body [11] (Durkee et al.,2019). Physical fitness refers to the strength, speed, endurance, agility, flexibility and other functions that the human body exhibits in its activities, physical quality is the external expression of a person's physical strength [12] (Hancox et al.,2018).

College students are in the final stage of moving from school to society, and the personality development and psychological development of college students during this period are extremely important for their future life and social development [13] (Hadler et al.,2021). If students can develop good and stable body self-esteem, it will not only improve overall self-esteem, but also have a positive impact on the individual's mental health [6] (Franzoi et al.,1984). College students' evaluation and perception of their body self-esteem is related to

their future physical activity awareness and physical life habits [14] (Deng et al.,2009).Aerobic exercise was advocated by Dr. Kenneth H. Cooper in 1968. Aerobic exercise is an exercise in which the energy consumed by the body during exercise is mainly supplied by the process of aerobic oxidation, usually lasting more than 30 minutes [15, 16] (Chen et al.,2011;Hung et al.,2021). Aerobic exercise can make the body release beta-endorphin, which can regulate the activity of the human brain nervous system, and it helps to relieve and eliminate depression, irritability, sadness, anger and other bad moods [17–19] (Li et al.,2003). A large number of studies have shown that different types of aerobic exercise can improve the mental health of college students and promote the formation of good mental qualities [4, 20–23] (Li et al.,2022;Kim et al.,2021;Zhang et al.,2022;Tang et al.,2022;Margulis et al.,2021). It has also been shown that aerobic exercise can increase the level of body self-esteem of college students [24–26] (He et al.,2002;He et al.,2003;Zhang et al.,2019 [18]). The effect of different aerobic exercises on improving the five dimensions of body self-esteem in college students varies, and different aerobic exercise on improving the five dimensions of body self-esteem among college students with different physical conditions and different genders are different, and there is a lack of quantitative systematic evaluation and research reports. Therefore, this study uses Meta-analysis to quantitatively summarize the recent literature and compare the effects of various aerobic exercises on the five dimensions of body self-esteem of college students of different genders and physical conditions, so as to provide reference for college physical education practice teaching and college students' exercise selection, and to improve the quality of college physical education and promote college students' mental health and exercise motivation.

## 2. Sources and methods

The process and steps of this study were conducted according to the regulations and process requirements of the Preferred Reporting Items for Systematic Reviews and Meta-analysis (PRISMA) [27]. The purpose of the PRISMA reputation is to help authors improve the writing and reporting of systematic reviews/meta-analyses(registration number CRD42023417686) (Table 1).

### 2.1 Information retrieval strategy

The literature was collected through PubMed, Web of Science, Cochrane Library, CNIK (Chinese National Knowledge Infrastructure), VIP Chinese Journal Service Platform, and WAN-FANG Data Knowledge Service Platform search platforms. The search terms were "aerobic exercise", "exercise intervention", "physical exercise", "university students", "Chinese university students", and "Chinese university students". "Meta-analysis," "body self-esteem," "Chinese college students," "physical exercise," "aerobic exercise," "exercise intervention," trying to retrieve each synonym and adjusting it to the specific database, the search time was built until May 2022, and the search method was a combination of subject terms, keywords The search method was a combination of subject terms, keywords, and article names.

### 2.2 Literature inclusion criteria

(1) The language of the included studies was English or Chinese, but the subjects were Chinese university students; (2) The intervention for the experimental group was a specific aerobic exercise program, while the experimental group must have a pre-test, the intervention duration was not less than 8 weeks, and the duration of each exercise session was at least 30 minutes (including 30 minutes); the control group took non-exercise exercise or a different exercise mode than the experimental group; (3) The body self-esteem measures were all developed

**Table 1. Reporting quality of included systematic reviews assessed using the PRISMA checklist.**

| Section and Topic | Item # | Checklist item | Location where item is reported |
|---|---|---|---|
| **TITLE** | | | |
| Title | 1 | Identify the report as a systematic review. | 1 |
| **ABSTRACT** | | | |
| Abstract | 2 | See the PRISMA 2020 for Abstracts checklist. | 12–37 |
| **INTRODUCTION** | | | |
| Rationale | 3 | Describe the rationale for the review in the context of existing knowledge. | 39–83 |
| Objectives | 4 | Provide an explicit statement of the objective(s) or question(s) the review addresses. | 84–93 |
| **METHODS** | | | |
| Eligibility criteria | 5 | Specify the inclusion and exclusion criteria for the review and how studies were grouped for the syntheses. | 117–125 |
| Information sources | 6 | Specify all databases, registers, websites, organisations, reference lists and other sources searched or consulted to identify studies. Specify the date when each source was last searched or consulted. | 107–115 |
| Search strategy | 7 | Present the full search strategies for all databases, registers and websites, including any filters and limits used. | 109–115 |
| Selection process | 8 | Specify the methods used to decide whether a study met the inclusion criteria of the review, including how many reviewers screened each record and each report retrieved, whether they worked independently, and if applicable, details of automation tools used in the process. | 123–125 |
| Data collection process | 9 | Specify the methods used to collect data from reports, including how many reviewers collected data from each report, whether they worked independently, any processes for obtaining or confirming data from study investigators, and if applicable, details of automation tools used in the process. | 15–23 |
| Data items | 10a | List and define all outcomes for which data were sought. Specify whether all results that were compatible with each outcome domain in each study were sought (e.g. for all measures, time points, analyses), and if not, the methods used to decide which results to collect. | 49–67 |
| | 10b | List and define all other variables for which data were sought (e.g. participant and intervention characteristics, funding sources). Describe any assumptions made about any missing or unclear information. | 131–137 |
| Study risk of bias assessment | 11 | Specify the methods used to assess risk of bias in the included studies, including details of the tool(s) used, how many reviewers assessed each study and whether they worked independently, and if applicable, details of automation tools used in the process. | 139–147 |
| Effect measures | 12 | Specify for each outcome the effect measure(s) (e.g. risk ratio, mean difference) used in the synthesis or presentation of results. | 149–155 |
| Synthesis methods | 13a | Describe the processes used to decide which studies were eligible for each synthesis (e.g. tabulating the study intervention characteristics and comparing against the planned groups for each synthesis (item #5)). | 161–164 |
| | 13b | Describe any methods required to prepare the data for presentation or synthesis, such as handling of missing summary statistics, or data conversions. | 166–170 |
| | 13c | Describe any methods used to tabulate or visually display results of individual studies and syntheses. | 145–151 |
| | 13d | Describe any methods used to synthesize results and provide a rationale for the choice(s). If meta-analysis was performed, describe the model(s), method(s) to identify the presence and extent of statistical heterogeneity, and software package(s) used. | 149–155 |
| | 13e | Describe any methods used to explore possible causes of heterogeneity among study results (e.g. subgroup analysis, meta-regression). | 187–192 201–205 214–218 227–231 240–244 |
| | 13f | Describe any sensitivity analyses conducted to assess robustness of the synthesized results. | 145–151 |
| Reporting bias assessment | 14 | Describe any methods used to assess risk of bias due to missing results in a synthesis (arising from reporting biases). | 193–197 206–210 219–223 232–236 245–249 |
| Certainty assessment | 15 | Describe any methods used to assess certainty (or confidence) in the body of evidence for an outcome. | 139–147 |

*(Continued)*

**Table 1.** (Continued)

| Section and Topic | Item # | Checklist item | Location where item is reported |
|---|---|---|---|
| **RESULTS** | | | |
| Study selection | 16a | Describe the results of the search and selection process, from the number of records identified in the search to the number of studies included in the review, ideally using a flow diagram. | 126–129 |
| | 16b | Cite studies that might appear to meet the inclusion criteria, but which were excluded, and explain why they were excluded. | 123–125 |
| Study characteristics | 17 | Cite each included study and present its characteristics. | 160–171 |
| Risk of bias in studies | 18 | Present assessments of risk of bias for each included study. | 173–176 |
| Results of individual studies | 19 | For all outcomes, present, for each study: (a) summary statistics for each group (where appropriate) and (b) an effect estimate and its precision (e.g. confidence/credible interval), ideally using structured tables or plots. | 160–164<br>187–192<br>201–205<br>214–218<br>227–231<br>240–244 |
| Results of syntheses | 20a | For each synthesis, briefly summarise the characteristics and risk of bias among contributing studies. | 364–374 |
| | 20b | Present results of all statistical syntheses conducted. If meta-analysis was done, present for each the summary estimate and its precision (e.g. confidence/credible interval) and measures of statistical heterogeneity. If comparing groups, describe the direction of the effect. | 176–188<br>197–201<br>210–214<br>223–227<br>236–240 |
| | 20c | Present results of all investigations of possible causes of heterogeneity among study results. | 187–192<br>201–205<br>214–218<br>227–231<br>240–244 |
| | 20d | Present results of all sensitivity analyses conducted to assess the robustness of the synthesized results. | 134–152 |
| Reporting biases | 21 | Present assessments of risk of bias due to missing results (arising from reporting biases) for each synthesis assessed. | 359–360 |
| Certainty of evidence | 22 | Present assessments of certainty (or confidence) in the body of evidence for each outcome assessed. | 135–143 |
| **DISCUSSION** | | | |
| Discussion | 23a | Provide a general interpretation of the results in the context of other evidence. | 250–357 |
| | 23b | Discuss any limitations of the evidence included in the review. | 358–366 |
| | 23c | Discuss any limitations of the review processes used. | N/A |
| | 23d | Discuss implications of the results for practice, policy, and future research. | 367–378 |
| **OTHER INFORMATION** | | | |
| Registration and protocol | 24a | Provide registration information for the review, including register name and registration number, or state that the review was not registered. | 98 |
| | 24b | Indicate where the review protocol can be accessed, or state that a protocol was not prepared. | N/A |
| | 24c | Describe and explain any amendments to information provided at registration or in the protocol. | N/A |
| Support | 25 | Describe sources of financial or non-financial support for the review, and the role of the funders or sponsors in the review. | 379–381 |
| Competing interests | 26 | Declare any competing interests of review authors. | 376–378 |
| Availability of data, code and other materials | 27 | Report which of the following are publicly available and where they can be found: template data collection forms; data extracted from included studies; data used for all analyses; analytic code; any other materials used in the review. | 382–383 |

using the Fox and the revised Physical Self-Esteem Scale (PSPP) for college students by Xu Xia and Yao Jiaxin [7] (Xu et al.,2001); (4) The outcome indicators must include five dimensions in physical self-esteem: physical self-worth, athletic ability, physical condition, physical attractiveness, and physical quality.(5) The included studies were experimental studies of exercise interventions.

## 2.3 Literature exclusion criteria

(1) experimental subjects were not Chinese university students, (2) incomplete outcome indicators, (3) review literature, (4) measurement tools did not meet the requirements (5) low quality literature and suspicious results (Fig 1).

## 2.4 Literature screening and data extraction

This literature exercise software EndNote X9 was used to help literature screening. The main contents of the extraction were: the first author of the collected literature, time of publication, experimental subjects, sample size, number of each gender, form of exercise, intervention period, frequency, single intervention time, and outcome evaluation index (some of the papers extracted in this study lacked control group data, so all the control group data were replaced with experimental group pre-test) [28]. The data included in the article are published and no unpublished data were included in the article.

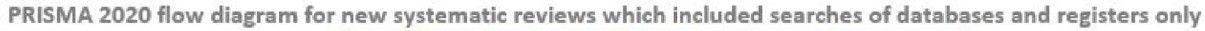

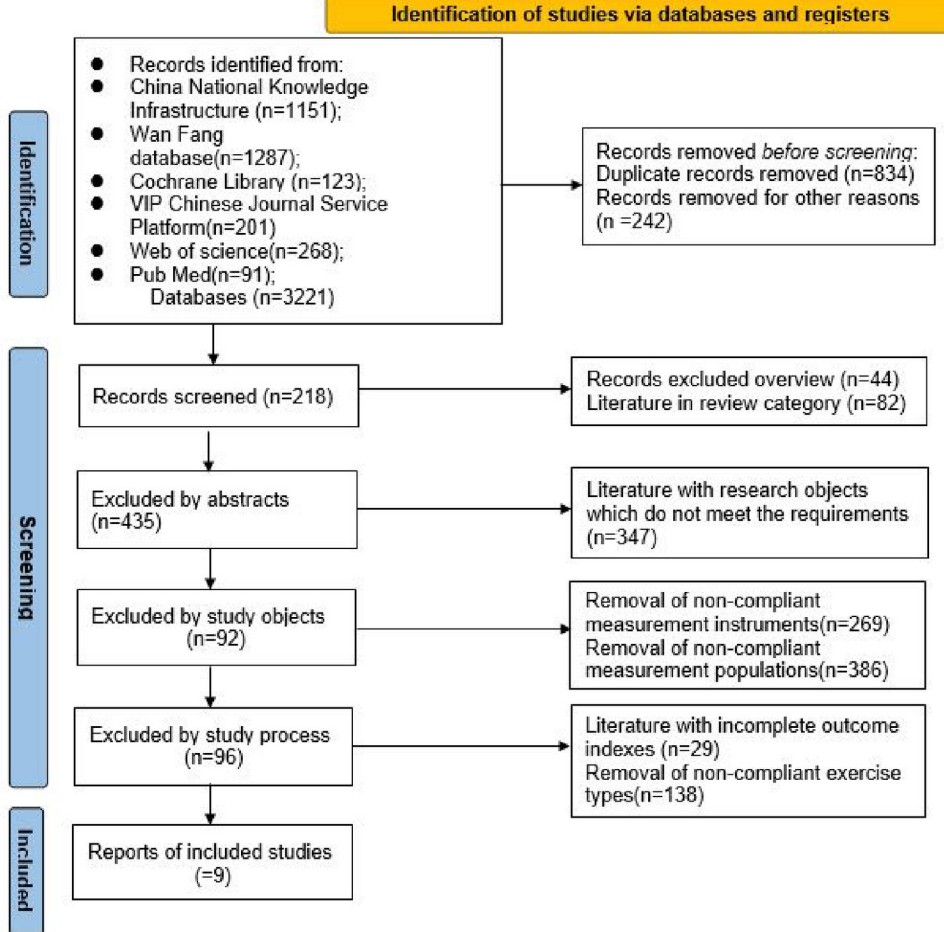

PRISMA 2020 flow diagram for new systematic reviews which included searches of databases and registers only

**Identification of studies via databases and registers**

**Identification**

- Records identified from:
- China National Knowledge Infrastructure (n=1151);
- Wan Fang database(n=1287);
- Cochrane Library (n=123);
- VIP Chinese Journal Service Platform(n=201)
- Web of science(n=268);
- Pub Med(n=91);
  Databases (n=3221)

Records removed *before screening*: Duplicate records removed (n=834) Records removed for other reasons (n =242)

**Screening**

Records screened (n=218)

Records excluded overview (n=44) Literature in review category (n=82)

Excluded by abstracts (n=435)

Literature with research objects which do not meet the requirements (n=347)

Excluded by study objects (n=92)

Removal of non-compliant measurement instruments(n=269) Removal of non-compliant measurement populations(n=386)

Excluded by study process (n=96)

Literature with incomplete outcome indexes (n=29) Removal of non-compliant exercise types(n=138)

**Included**

Reports of included studies (=9)

*Consider, if feasible to do so, reporting the number of records identified from each database or register searched (rather than the total number across all databases/registers).

**If automation tools were used, indicate how many records were excluded by a human and how many were excluded by automation tools.

From: Page MJ, McKenzie JE, Bossuyt PM, Boutron I, Hoffmann TC, Mulrow CD, et al. The PRISMA 2020 statement: an updated guideline for reporting systematic reviews. BMJ 2021;372:n71. doi: 10.1136/bmj.n71 For more information, visit: http://www.prisma-statement.org/

**Fig 1. Preferred Reporting Items for Systematic Reviews and Meta-Analyses (PRISMA) flow diagram.**

## 2.5 Quality evaluation criteria

Quality evaluation was conducted independently by two researchers. Grade pro software was used to evaluate the quality of the outcome indicators in the literature. The evaluation included (1) study limitations-assessing whether the studies had limitations in design or execution serious enough to reduce the quality of evidence for that outcome; (2) inconsistency-assessing whether the results were consistent across studies and whether any inconsistencies were serious enough to reduce the quality of evidence for that outcome; (3) Imprecision-assessing whether the results are sufficiently precise and whether any imprecision in the results is serious enough to reduce the quality of evidence for that result. Systematic review authors and guideline panels define imprecision differently; (4) Indirectness-Assess whether the evidence directly answers the question posed, and whether the indirectness of the available evidence is serious enough to reduce the quality of the evidence for that outcome; (5) Publication bias-assesses whether publication bias is likely to exist and whether reporting bias is serious enough to reduce the quality of evidence for that outcome. Each outcome was categorised as "no", "serious" or "very serious".

## 2.6 Risk of literature bias assessment

The included literature was evaluated for risk of bias using Cochrane in Review Manager 5.4 software [29] (Higgins et al.,2011), and the quality of the included studies was evaluated for seven indicators: random sequence generation (selection bias), Allocation concealment (selection bias), Blinding of participants and personnel (performance bias), Blinding of outcome assessment (detection bias), Incomplete outcome data (attrition bias). Selgative reporting (reporting blas), Other bias, Evaluation results include low risk, uncertainty, and high risk. All evaluation indicators were satisfied as grade A, with the lowest possibility of bias; partially satisfied as grade B, with low possibility of bias; and completely unsatisfied as grade C, with high possibility of bias. The quality assessment was carried out independently by 2 sports psychologists.

## 2.7 Data processing

Excel software was used to organize and summarize the data, Review Manager 5.4 software was used to assess the risk of bias in the nine papers, StataMP17.0 software was used to test for heterogeneity, merge the data, and draw funnel plots and forest plots, and Random-effects Model (REM) was used for Meta-analysis. All nine papers included in this study had continuous variables and consistent units, so Weighted mean difference (WMD) was used as the effect indicator. Each effect size was expressed as 95% CI (confidence interval), and the consistency coefficient Q and $I^2$ values were used for heterogeneity analysis.

## 3. Results

### 3.1 Literature search results

The initial literature search of 3221 articles was conducted, and a total of 9 articles were finally included in the Meta-analysis after screening.

### 3.2 Basic characteristics of the included studies

As shown in Table 2, of the nine included papers [30–38] (Zhu et al.,2005;Zhu et al.,2006; Wang.,2016;Yuan.,2021;Wang et al.,2012;Li.,2007;Cao et al.,2016;Xu et al.,2020;Sun et al.,2018), the subjects included 1613 individuals.

**Table 2. Summary of studies that met the inclusion criteria.**

| First Author | Publish time | | Male / Female | Exercise Program | Exercise frequency | Movement time/min | Heart rate | Duration of intervention |
|---|---|---|---|---|---|---|---|---|
| Xu Hengzheng | 2020 | 30 | Not explicitly stated | Ba duan jin | 3 times/week | 40~60 | 100~130 | 10 weeks |
| Wenshu Sun | 2018 | 500 | 291/209 | Wushu+Aerobics+Physical Dance+Badminton +Table tennis+Tennis+Football+Basketball | 1 time/week | 90 | 120~140 | One academic year |
| Beijuan Cao | 2016 | 184 | 0/184 | Aerobics+Yoga+Sports Dance+Artistic Gymnastics | 1 time/week | 90 | 110~140 | One semester |
| Li Lingyun | 2007 | 255 | 137/118 | Taijiquan | 1 time/week | 90 | 100~120 | 16 weeks |
| Wang Aijing | 2012 | 186 | 112/74 | Aerobic running + sports games + light equipment strength training | 3 times/week | 90 | 100~140 | 15 weeks |
| Yuan Yuxin | 2021 | 60 | 0/60 | Physical Dance | 2 times/week | 90 | 120~150 | 16 weeks |
| Wang Yajun | 2016 | 195 | 88/107 | Jogging+self weight training | 3 times/week | 50 | 110~150 | 12 weeks |
| Zhu Fengshu | 2006 | 87 | 0/87 | small-intensity aerobics+ medium-intensity aerobics | 3 times/week | 30 | 90~150 | 10 weeks |
| Zhu Fengshu | 2005 | 116 | 57/59 | Small to medium intensity basketball and aerobics | 3 times/week | 30 | 90~150 | 10 weeks |

In the nine papers, some researchers used different exercise programs to intervene in the body self-esteem of college students with different characteristics, so all the experimental groups of the intervention of exercise programs for college students with different characteristics in the nine included papers were extracted, and a total of 28 experimental groups were extracted for Meta-analysis, as shown in Table 3.

### 3.3 Risk of bias evaluation

Nine included papers were evaluated for quality and risk of bias according to the Cochrane Risk of Bias Assessment Tool, of which one was of low quality and high risk, and the remaining eight were of moderate quality(Fig 2).

### 3.4 GRADE system recommended grading

The evidence obtained on the effects of aerobic exercise on the physical self-esteem of Chinese university students was graded according to the GRADE system, and their quality of evidence was low, respectively (Table 4).

### 3.5 Meta-analysis results

**3.5.1 Heterogeneity test and publication bias analysis of the effect of aerobic exercise workout on physical self-worth of college students.** *3.5.1.1 Heterogeneity and combined effect value test.* Forest plots in Meta-analysis are graphs based on statistical effect sizes and confidence intervals, using the results of numerical operations included in the literature, and thus plotted [39, 40] (Dettori et al.,2021). Bias in the literature is also called systematic error, which refers to the process by which the results or inferences of the final study deviate from the true value or lead to the phenomenon of such deviation. It is any tendency that can lead to the final conclusion systematically differing from the true value, generally during the process of collecting information, analyzing data, or publishing [41–43]. The numerical results yielded high heterogeneity among the studies: $Q = 89.94, df = 27$ ($p<0.01$), $I^2 = 70\%$, and according to the criteria for classifying heterogeneity in the literature [44] (Hatala et al.,2002), a random effects model was used to test for heterogeneity and combined effect values for 28 groups, (see Fig 1). ($WMD = 1.46$, 95% $CI$: 1.08 to 1.83, $p < 0.001$), which was statistically significant. It

**Table 3. The 28 experimental groups included in the meta-analysis.**

| Experimental groups | Number of persons |
|---|---|
| Xu Hengzheng 2020 (Baduanjin) | 30 |
| Sun Wenshu 2018 (Wushu) | 78 |
| Sun Wenshu 2018 (Aerobics) | 41 |
| Sun Wenshu 2018 (Sports Dance) | 39 |
| Sun Wenshu 2018 (Table Tennis) | 87 |
| Sun Wenshu 2018 (Badminton) | 70 |
| Sun Wenshu 2018 (Tennis) | 37 |
| Sun Wenshu 2018 (Basketball) | 73 |
| Sun Wenshu 2018 (Soccer) | 75 |
| Beijuan Cao 2016 (Aerobics) | 23 |
| Beijuan Cao 2016 (Yoga) | 23 |
| Beijuan Cao 2016 (Sports Dance) | 23 |
| Beijuan Cao 2016 (Artistic Gymnastics) | 23 |
| Li Lingyun 2007 (Taijiquan) | 255 |
| Wang Aijing 2012 (Aerobic exercise) Low body mass male | 31 |
| Wang Aijing 2012 (aerobic exercise) low body mass female | 21 |
| Wang Aijing 2012 (Aerobic exercise) Super fit male | 52 |
| Wang Aijing 2012 (aerobic exercise) super fit female | 36 |
| Wang Aijing 2012 (Aerobic Exercise) Obesity Male | 28 |
| Wang Aijing 2012 (aerobic exercise) obese female | 18 |
| Yuan Yuxin 2021 (physical dance) | 30 |
| Wang Yajun 2016 (jogging + self weight training) | 99 |
| Zhu Fengshu 2006 (small-intensity aerobics) | 28 |
| Zhu Fengshu 2006 (medium-intensity aerobics) | 29 |
| Zhu Fengshu 2005 (small intensity basketball) Male | 28 |
| Zhu Fengshu 2005 (medium-intensity basketball) Male | 29 |
| Zhu Fengshu 2005 (small-intensity aerobics) female | 29 |
| Zhu Fengshu 2005 (medium-intensity aerobics) female | 30 |

indicates that aerobic exercise can effectively improve the physical self-worth of Chinese university students. (Fig 3). Subgroup analyses were conducted by gender for each study, with statistically significant differences between subgroups ($p < 0.05$, $I^2 > 50\%$), and meta-analyses were conducted using a random-effects model. The results showed that the sense of physical self-worth of each study was higher than the preexperimental measure ($p < 0.05$). As shown in Table 5.

*3.5.1.2 Bias analysis of the research literature.* As shown in Fig 4, the 28 experimental groups were distributed on both sides of the funnel plot, with most of them within the two dashed 95% CI lines and four groups outside the 95% CI line, which may have heterogeneity. The overall quality of the study literature was moderate, and there was no significant bias among the literature (Fig 4).

**3.5.2 Heterogeneity test and publication bias analysis of the effect of aerobic exercise workout on college students' Athletic ability.** *3.5.2.1 Tests for heterogeneity and combined effect values.* The results of numerical operations yielded a high heterogeneity among studies: $Q = 74.99$, $df = 27$ ($p<0.01$), $I^2 = 64\%$, and a random effects model was used to test for heterogeneity and combined effect values for the 28 experimental groups. ($WMD = 1.62$, 95% $CI$: 1.23 to 2.01, $p<0.001$), which was statistically significant. It showed that aerobic exercise was effective in improving the Athletic ability of Chinese college students, and there was no

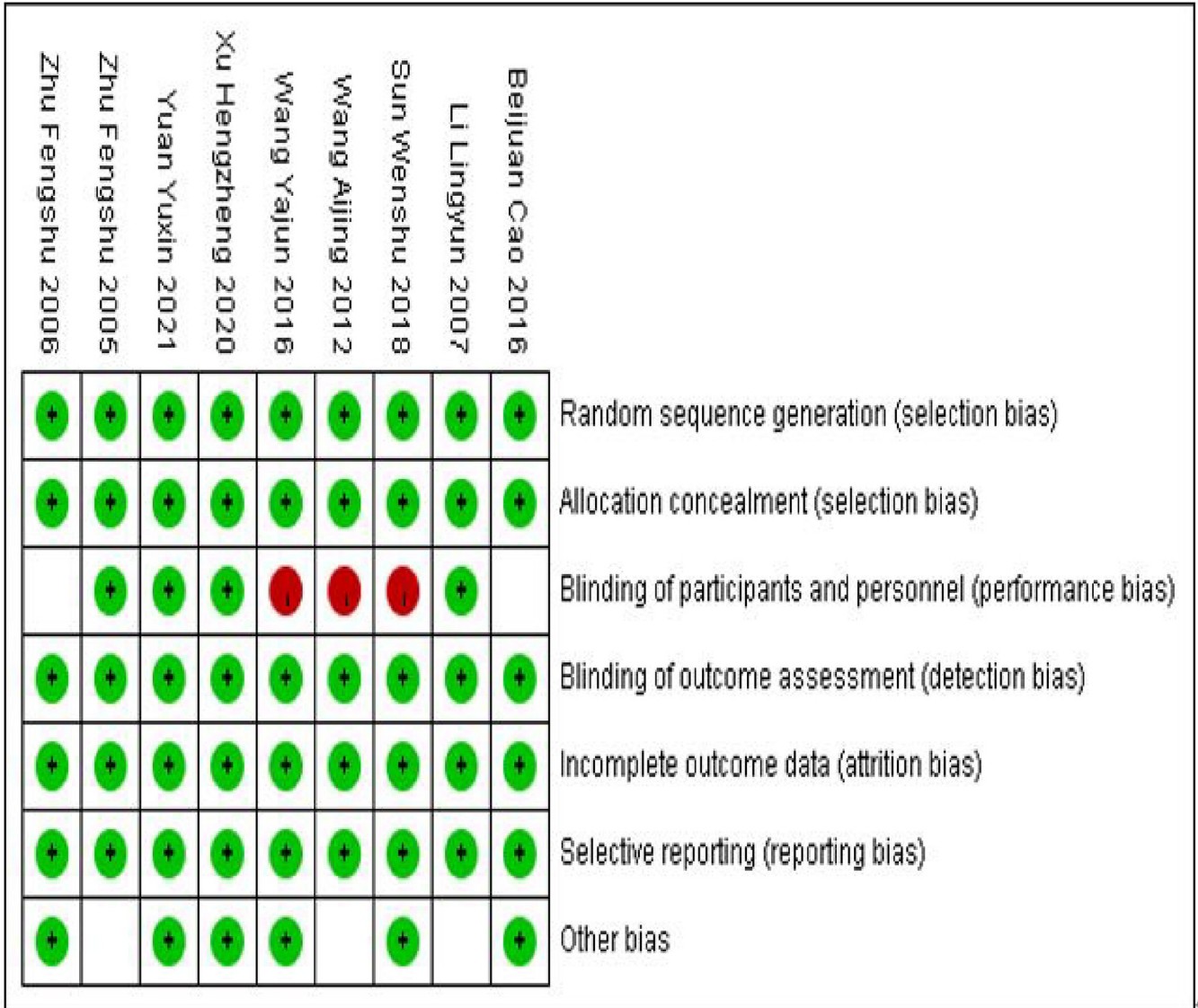

**Fig 2. Risk of bias summary: Review authors' judgements about each risk of bias item for each included study.**

significant bias in the literature, (Fig 5). The studies were analysed in subgroups by gender, with greater heterogeneity occurring within aerobic exercise intervention females ($I^2 > 50\%$), and meta-analysis was performed using a random effects model. There was little heterogeneity in aerobic exercise interventions in women ($p < 0.00001$, $I^2 = 2\%$), and exercise capacity was higher than preexperimental measurements in all studies ($p < 0.05$). As shown in Table 6.

*3.5.2.2 Analysis of bias in the research literature.* As shown in (Fig 6), the 28 experimental groups were distributed on both sides of the funnel plot, and most of them were within the two dashed 95% CI lines, with three groups outside the 95% CI line, which may have heterogeneity. The overall quality of the study literature was moderate, and there was no significant bias among the literature.

**3.5.3 Heterogeneity test and publication bias analysis of the effect of aerobic exercise workouts on college students' physical status.** *3.5.3.1 Tests for heterogeneity and combined*

**Table 4. GRADE system recommended grading results.**

**Aerobic Exercise for [Body Self-Esteem] in [Chinese College Students]**

Patient or population:
Settings:
Intervention: Chinese College Students
Comparison:

| Outcomes | Illustrative comparative risks* (95% CI) | | Relative effect (95% CI) | No of Participants (studies) | Quality of the evidence (GRADE) | Comments |
|---|---|---|---|---|---|---|
| | Assumed risk | Corresponding risk | | | | |
| | | **Chinese College Students** | | | | |
| physical self-worth | | The mean physical self-worth in the intervention groups was **1.46 higher** (1.08 to 1.83 higher) | | 2730 (9 studies) | ⊕⊕⊖⊖ **low**[1,2] | |
| exercise capacity | | The mean exercise capacity in the intervention groups was **1.62 higher** (1.23 to 2.01 higher) | | 2730 (9 studies) | ⊕⊕⊖⊖ **low**[1,2] | |
| physical condition | | The mean physical condition in the intervention groups was **1.32 lower** (0.96 to 1.67 higher) | | 2730 (9 studies) | ⊕⊕⊖⊖ **low**[1,2] | |
| physical attractiveness | | The mean physical attractiveness in the intervention groups was **1.32 higher** (0.99 to 1.65 higher) | | 2730 (9 studies) | ⊕⊕⊖⊖ **low**[1,2] | |
| physical fitness | | The mean physical fitness in the intervention groups was **1.51 higher** (1.07 to 1.95 higher) | | 2730 (9 studies) | ⊕⊕⊖⊖ **low**[1,2] | |

*The basis for the **assumed risk** (e.g. the median control group risk across studies) is provided in footnotes. The **corresponding risk** (and its 95% confidence interval) is based on the assumed risk in the comparison group and the **relative effect** of the intervention (and its 95% CI).
**CI:** Confidence interval;

GRADE Working Group grades of evidence
**High quality:** Further research is very unlikely to change our confidence in the estimate of effect.
**Moderate quality:** Further research is likely to have an important impact on our confidence in the estimate of effect and may change the estimate.
**Low quality:** Further research is very likely to have an important impact on our confidence in the estimate of effect and is likely to change the estimate.
**Very low quality:** We are very uncertain about the estimate.

[1] Inclusion of studies biased in terms of blinding

[2] Heterogeneity exists

*effect values.* The results of numerical operations yielded high heterogeneity among studies: $Q = 58.22$, $df = 27$ ($p<0.01$), $I^2 = 53.6\%$, and a random effects model was used to test for heterogeneity and combined effect values for the 28 experimental groups. (*WMD* = 1.32, 95% *CI*: 0.98 to 1.67, $p<0.001$), which was statistically significant. It showed that aerobic exercise was effective in improving the physical status of Chinese college students and there was no significant bias in the literature, (Fig 7). The studies were analysed in subgroups by gender, with statistically significant and low heterogeneity of differences in each subgroup ($I^2 < 50\%$, $p < 0.05$), and meta-analysis was performed using a fixed-effects model. The results showed that the physical condition of each study participant was higher than the preexperimental measurements ($p < 0.05$). As shown in Table 7.

*3.5.3.2 Analysis of bias in the research literature.* The 28 experimental groups were distributed on both sides of the funnel plot, and most of them were within the two dashed 95% CI lines, with five groups outside the 95% CI line, which may have heterogeneity. The overall

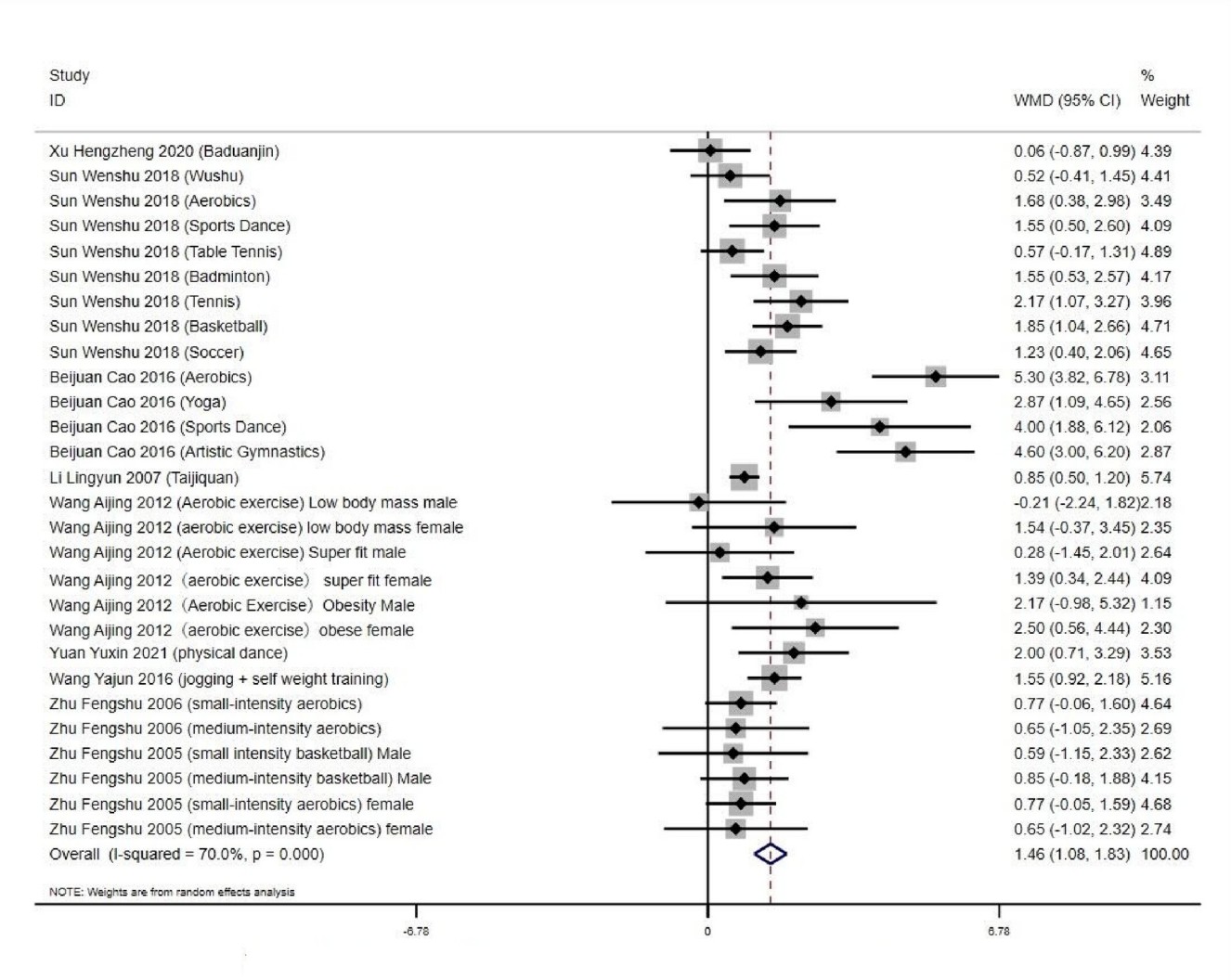

**Fig 3. Effect of aerobic exercise workout on physical self-worth of college students.**

quality of the study literature was moderate, and there was no significant bias among the literature, as shown in (Fig 8).

**3.5.4 Heterogeneity test and publication bias analysis of the effect of aerobic exercise workout on physical attractiveness of college students.** *3.5.4.1 Tests for heterogeneity and combined effect values.* The results of numerical operations yielded high heterogeneity among studies: $Q = 62.23$, $df = 27$ ($p<0.01$), $I^2 = 56.6\%$, and a random effects model was used to test for heterogeneity and combined effect values for the 28 experimental groups. ($WMD = 1.32$, 95% CI: 0.99 to 1.65, $p<0.001$), which was statistically significant. It showed that aerobic

**Table 5. Results of sub group analysis of body self-worth.**

| Subgroup (Male/Female) | Number of studies | MD (95%CI) | P | Z | $I^2\%$ |
|---|---|---|---|---|---|
| Male | 8 | -2.09 [-3.18, -1.00] | P = 0.0002 | 3.77 | 86% |
| Female | 8 | -1.58 [-2.36, -0.80] | P<0.0001 | 3.98 | 59% |

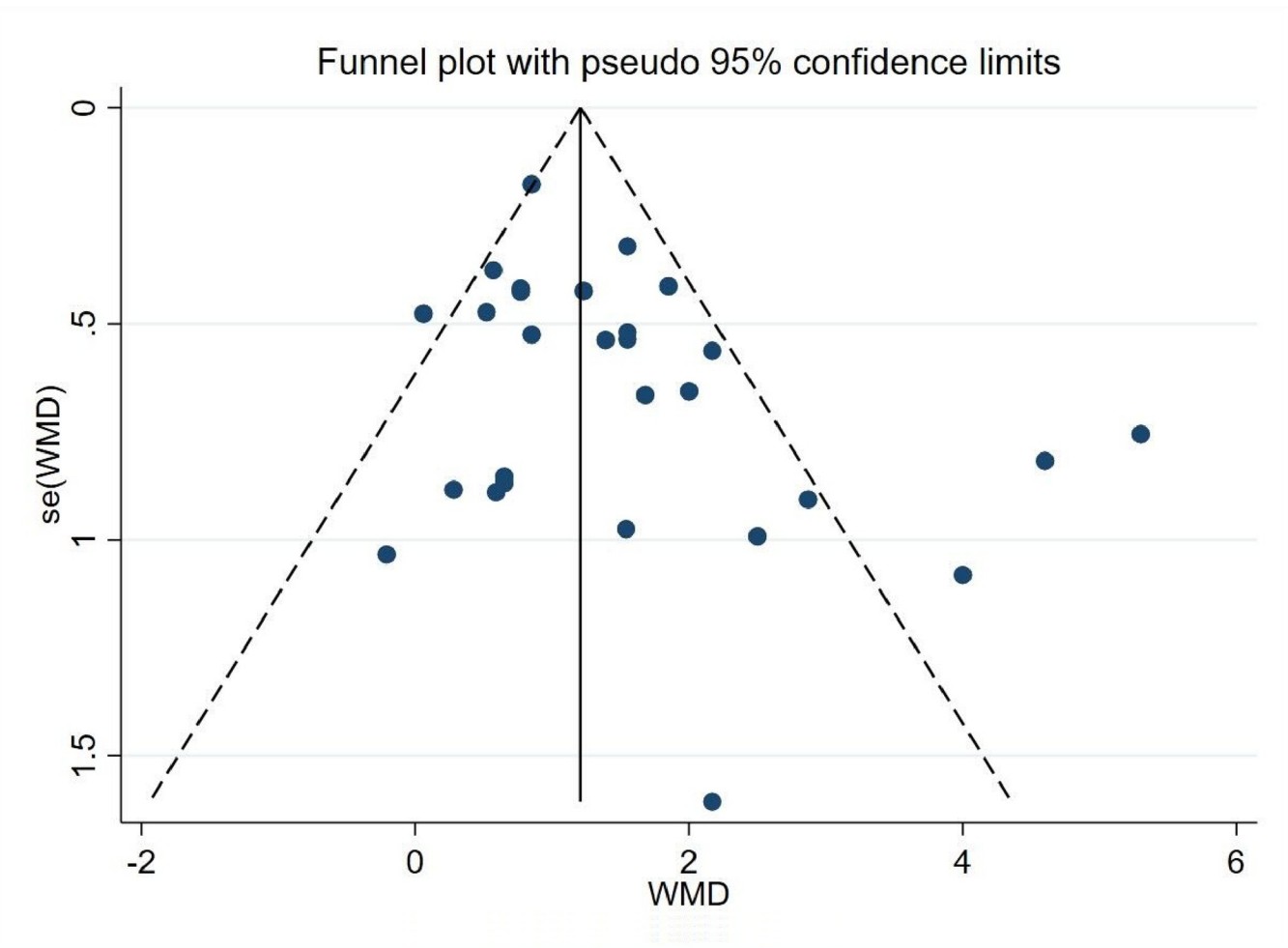

**Fig 4. Body self-worth funnel diagram.**

exercise was effective in improving the physical attractiveness of Chinese college students, and there was no significant bias in the literature, (Fig 9). The study was divided into gender subgroups, with statistically significant differences and low heterogeneity among subgroups (I2 < 50%, p < 0.05). A fixed effect model was used for meta-analysis. The results showed that the physical attractiveness of female study participants was higher than that measured before the experiment (p < 0.05), and the physical attractiveness of male study participants was not significantly improved (P = 0.21). As shown in Table 8.

*3.5.4.2 Analysis of bias in the research literature.* The 28 experimental groups were distributed on both sides of the funnel plot, and most of them were within the two dashed 95% CI lines, with four groups outside the 95% CI line, which may have heterogeneity. The overall quality of the study literature was moderate, and there was no significant bias among the literature, as shown in (Fig 10).

### 3.5.5 Heterogeneity test and publication bias analysis of the effect of aerobic exercise workouts on college students' physical fitness

*3.5.5.1 Tests for heterogeneity and combined effect values.* The results of numerical operations yielded a high heterogeneity among studies: $Q = 92.85$, $df = 27$ ($p<0.01$), $I^2 = 70.9\%$, and a

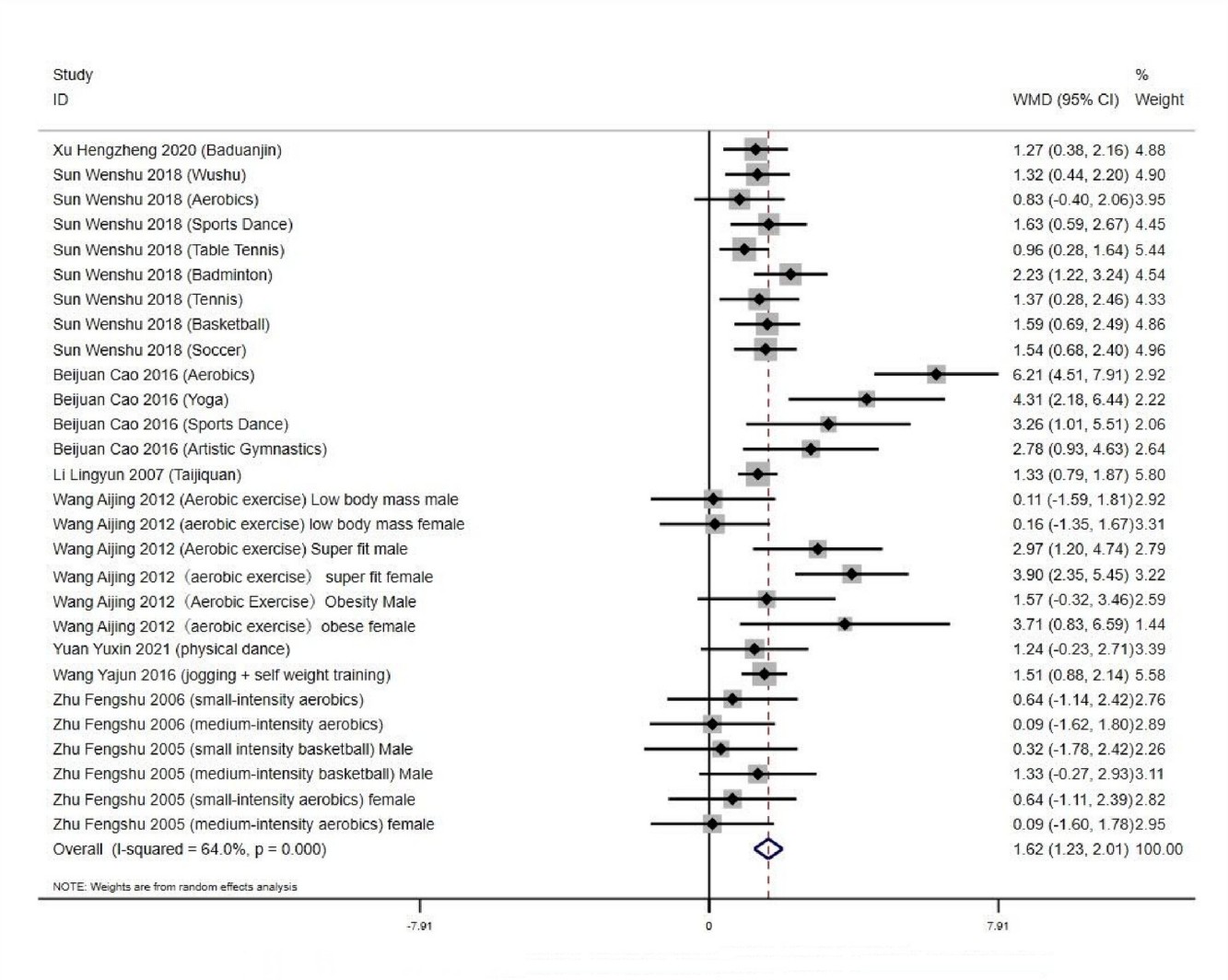

**Fig 5. Effect of aerobic exercise workout on exercise capacity of college students.**

random effects model was used to test for heterogeneity and combined effect values for the 28 experimental groups. ($WMD = 1.51$, 95% $CI$: 1.07 to 1.95, $p<0.001$), which was statistically significant. It indicates that aerobic exercise can effectively improve the physical fitness of Chinese college students, and there is no significant bias in the literature, (Fig 11). The study was subgroup analyzed by sex, with greater heterogeneity among men ($I^2 > 50\%$) with aerobic exercise intervention, and a random effects model was used for meta-analysis. There was little heterogeneity in aerobic exercise interventions in women ($p < 0.0001$, $I^2 = 48\%$), and exercise capacity was higher in all studies than in pre-experimental measurements ($p < 0.05$). As shown in Table 9.

**Table 6. Results of subgroup analysis of exercise capacity.**

| Subgroup (Male/Female) | Number of studies | MD (95%CI) | P | Z | $I^2\%$ |
|---|---|---|---|---|---|
| Male | 8 | -1.40 [-1.80, -1.01] | P<0.00001 | 6.99 | 2% |
| Female | 8 | -1.52 [-2.35, -0.70] | P = 0.0003 | 3.63 | 68% |

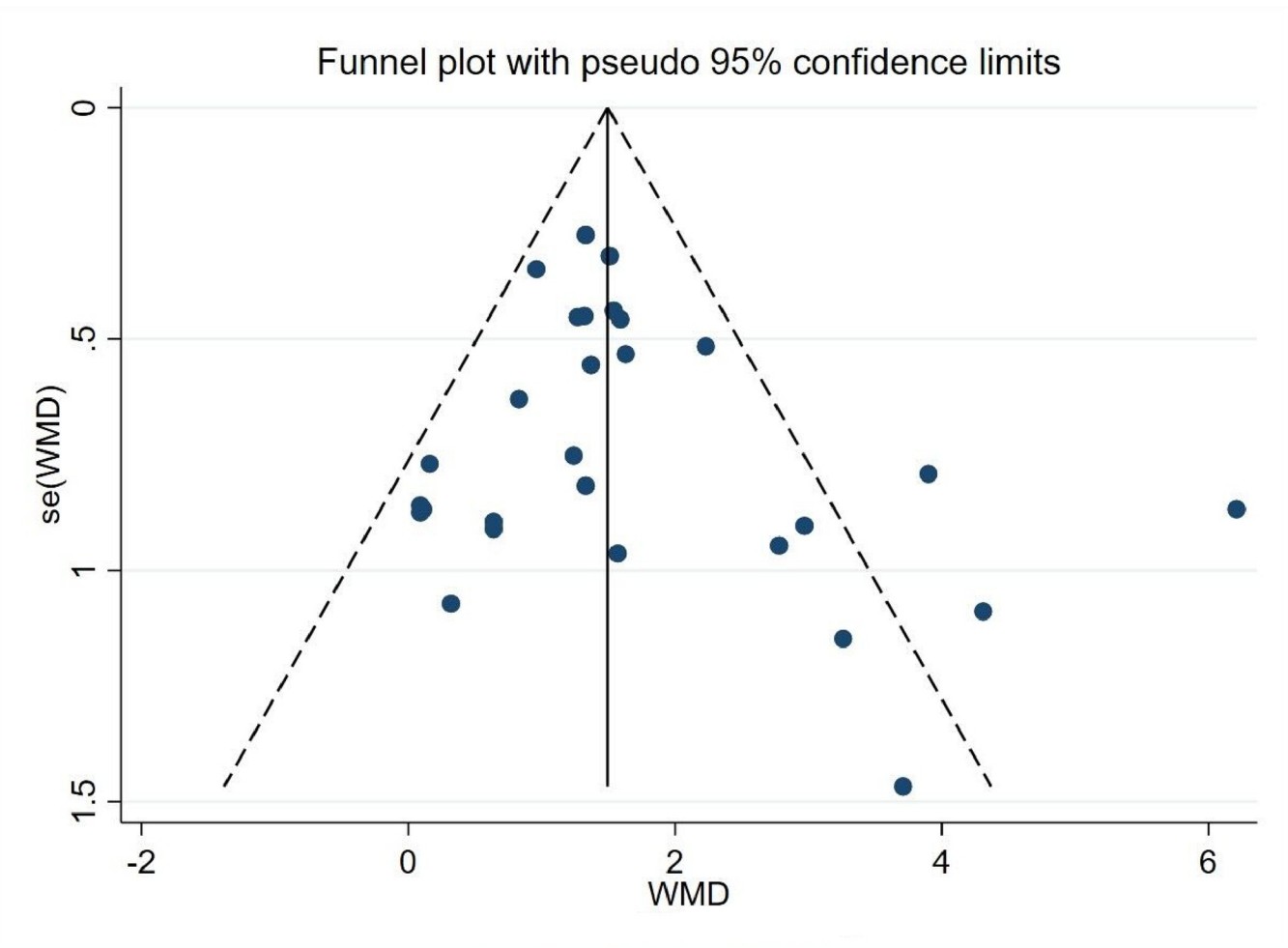

**Fig 6. Athletic ability funnel diagram.**

*3.5.5.2 Analysis of bias in the research literature.* The 28 experimental groups were distributed on both sides of the funnel plot, and most of them were within the two dashed 95% CI lines, with five groups outside the 95% CI line, which may have heterogeneity. The overall quality of the study literature was moderate, and there was no significant bias among the literature, as shown in (Fig 12).

# 4. Analysis and discussion

## 4.1 The effect of aerobic exercise on body self-esteem intervention of college students of different genders

The results of the subgroup analyses showed that male university students had higher levels of body self-esteem and higher dimensions of body self-esteem than female university students in both the pre- and post-experiment periods. This may be because Chinese female college students are pursuing unhealthy beauty, such as "chopstick legs", "A4 waist", "right-angle shoulders", and evaluating their beauty by thinness and whiteness, while exercise will lead to the development of human muscles, and if they choose to be outdoors, it is inevitable that their skin will become darker, so the level of body self-awareness is low, resulting in a lower level of

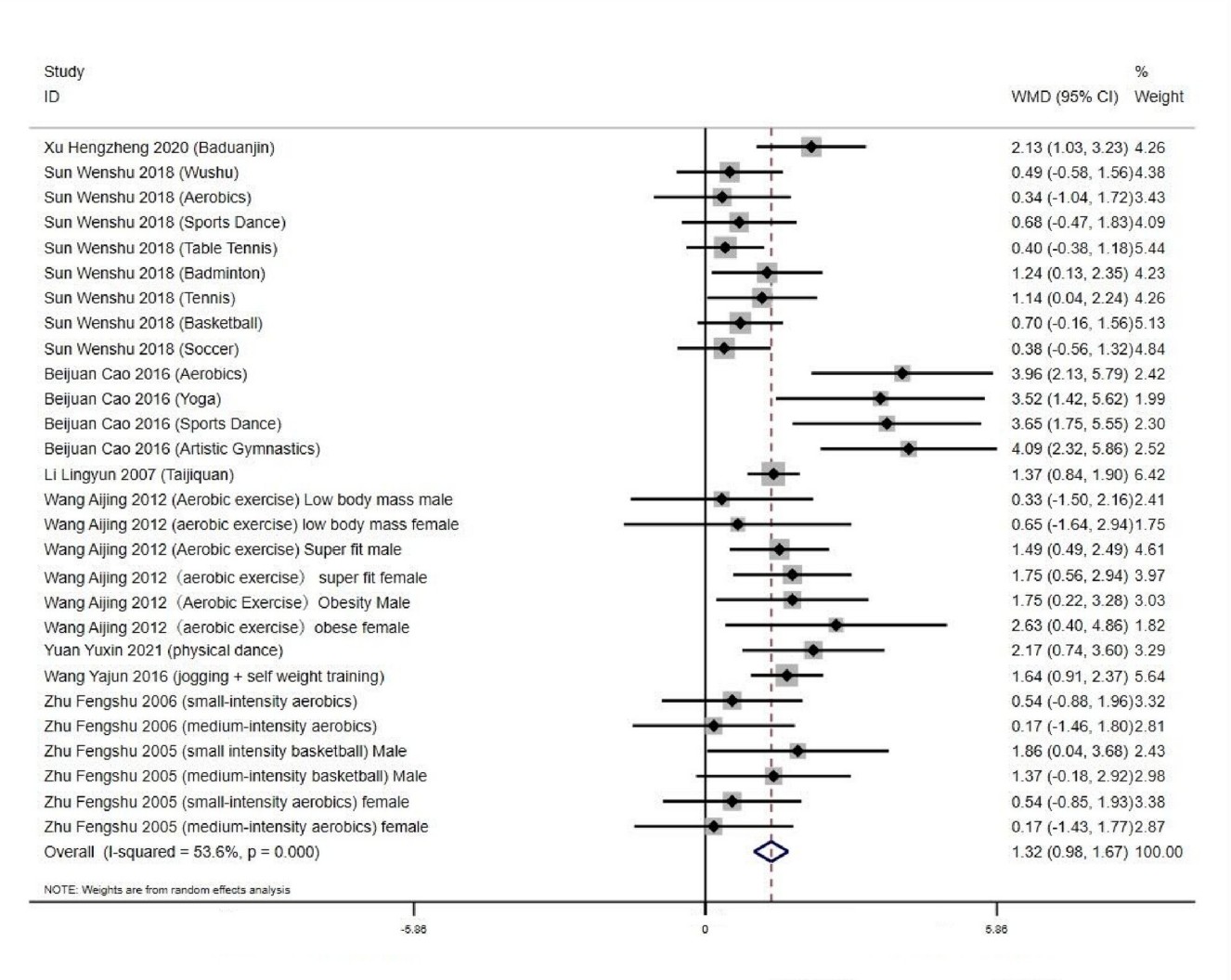

**Fig 7. Effect of aerobic exercise workout on the physical condition of college students.**

body self-esteem. In contrast, Chinese male college students exercise more frequently compared to female college students, and the ratio of men to women in college sports fields and gyms is basically greater for men than for women, while Chinese male college students do not pursue as much as Chinese female college students in terms of body perception [45] (Zhao et al.,2022), so the level of body self-esteem before the experiment is higher than that of female students. The intervention effects of physical attractiveness and physical self-worth were most significant for female college students after the experiment, followed by athletic ability, and finally physical condition and physical quality; the intervention effects of athletic ability were

**Table 7. Results of subgroup analysis of physical condition.**

| Subgroup (Male/Female) | Number of studies | MD (95%CI) | P | Z | I²% |
|---|---|---|---|---|---|
| Male | 8 | -0.83 [-1.25, -0.42] | P<0.0001 | 4.19 | 13% |
| Female | 8 | -0.99 [-1.43, -0.54] | P<0.0001 | 4.36 | 0% |

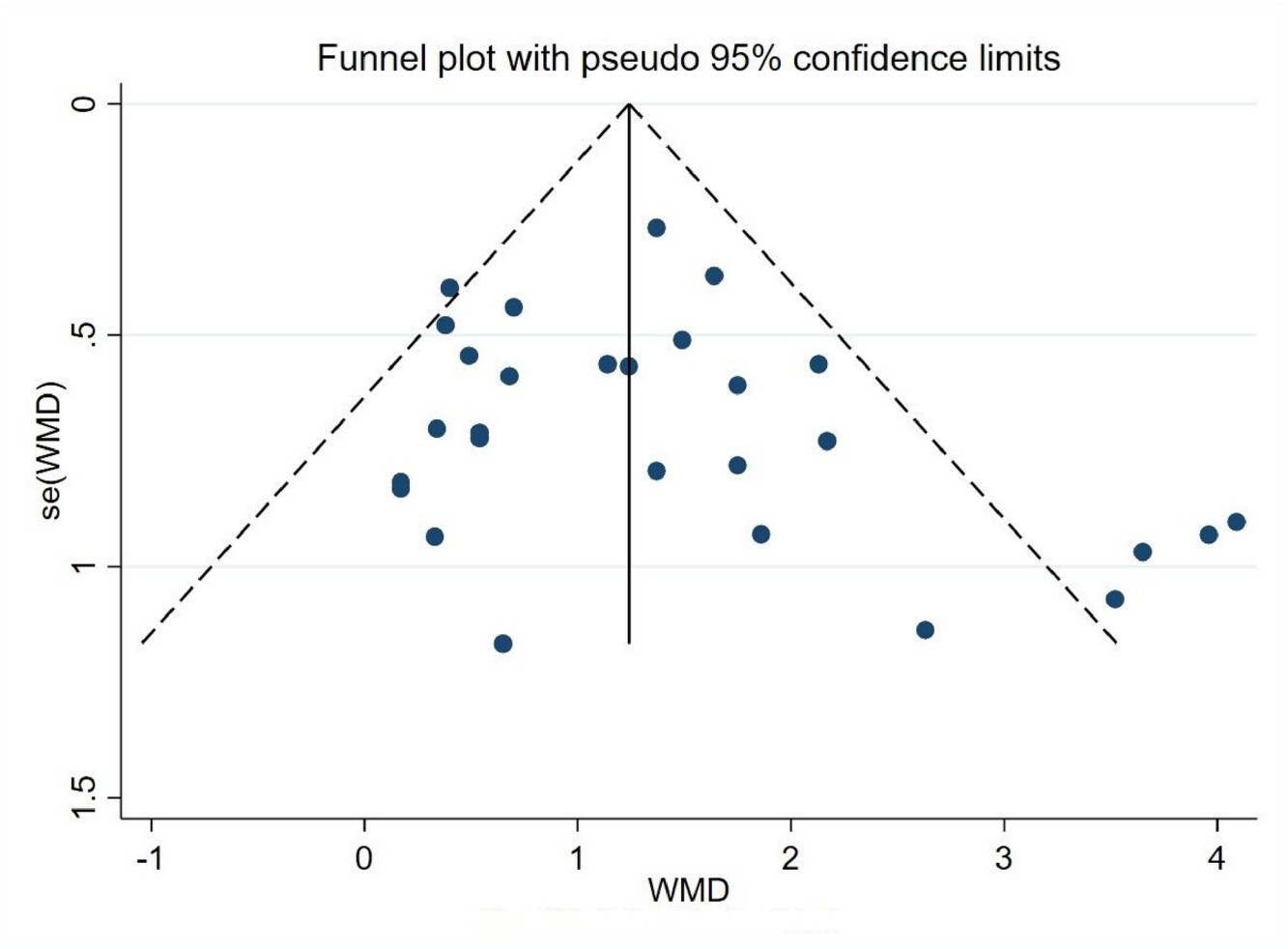

**Fig 8. Physical condition funnel chart.**

most significant for male college students, followed by general physical self-worth and physical quality, again physical attractiveness, and finally physical condition. This shows that the body self-esteem pursued by male and female students is different. Boys pursue is athletic ability, girls pursue is the sense of physical self-worth and physical attractiveness, which shows that boys and girls in sports exercise different exercise motivation and different exercise goals.

## 4.2 Effects of aerobic exercise workouts on body self-esteem interventions among college students with different physical fitness levels

In the nine included papers, the exercise interventions were targeted at college students with specific body types, including: overweight (BMI of 24–27.9), obese (BMI $\geq$ 28), low body type (BMI $<$ 18.5), and "weak" college students. From the data included in the literature, comparing the level of body self-esteem of this group of university students with special physique before the experiment with the other groups, the physical self-worth (mean of special physique group: 12.24; mean of other groups: 13.57), athletic ability (mean of special physique group: 12.55; mean of other groups: 13.39), physical condition (mean of special physique group: 13.76. Other group mean: 14.95), physical attractiveness (special physique group mean: 13.05; other group mean: 13.76), and physical fitness (special physique group mean: 11.65; other group

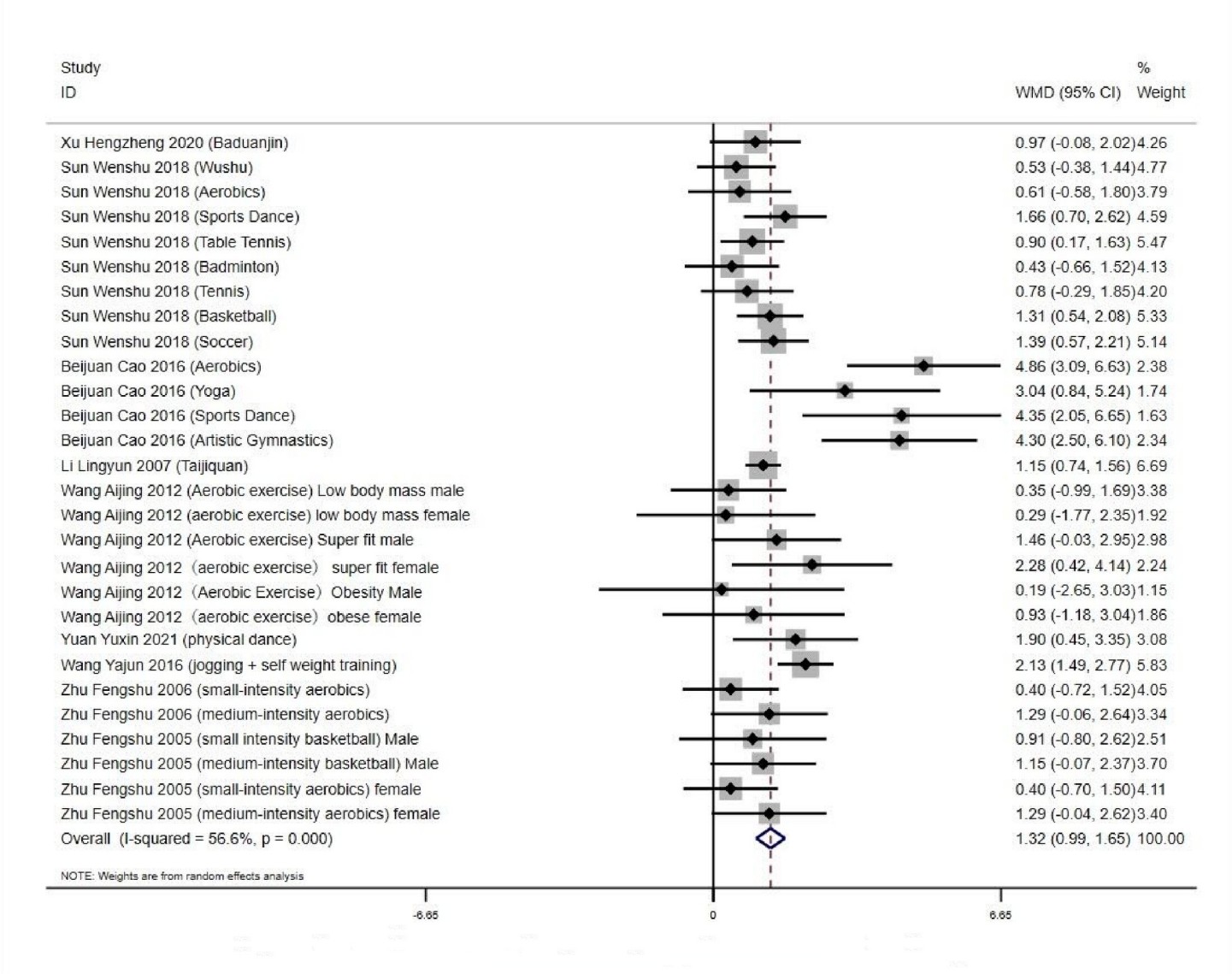

**Fig 9. Effect of aerobic exercise workout on physical attractiveness of college students.**

mean: 14.11). It can be seen that body self-esteem of college students with obese or thin bodies is significantly lower than other groups. In particular, the difference between physical quality and physical self-worth is the most obvious, but the level of body self-esteem of "weak" college students and low-physique college students is a little higher than that of obese and super-physique college students before the experiment, and related studies also show that there are obvious problems in the mental health of obese college students [46, 47] (Qiao.,2008;Wu.,2011),

**Table 8. Results of subgroup analysis of physical attractiveness.**

| Subgroup (Male/Female) | Number of studies | MD (95%CI) | P | Z | $I^2$% |
|---|---|---|---|---|---|
| Male | 8 | -0.33 [-0.86, 0.19] | P = 0.21 | 1.24 | 41% |
| Female | 8 | -1.02 [-1.42, -0.62] | P<0.00001 | 5.01 | 0% |

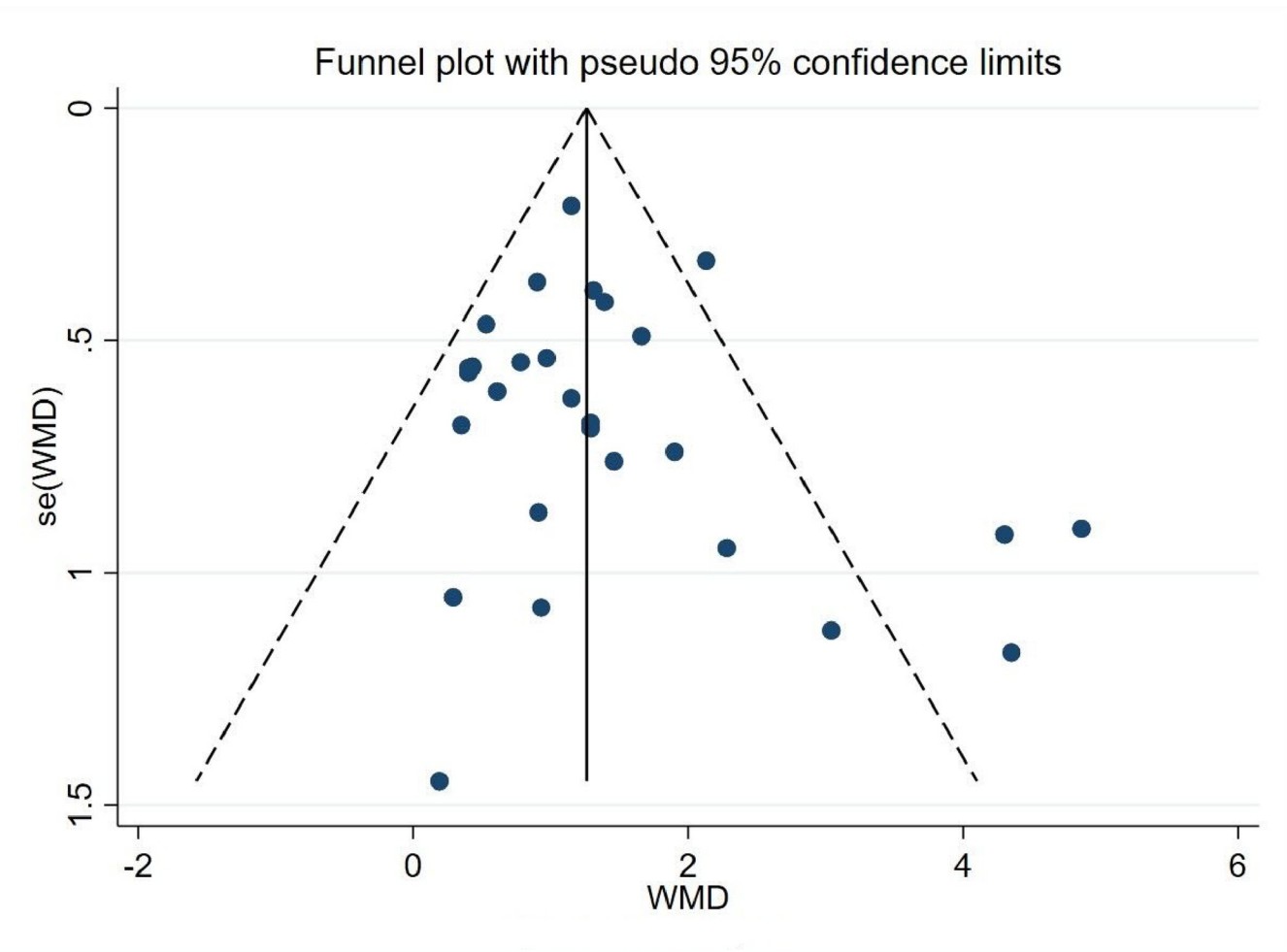

**Fig 10. Funnel diagram.**

and the main psychological problems are low self-esteem, lack of self-confidence, loneliness and depression. lack of self-confidence, loneliness and depression. After the intervention, the changes in body self-esteem were more pronounced in the hyperthymic and obese groups than in the low-body mass group and the "weak" college students. This may be due to the fact that the accumulation of body fat in the overweight and obese groups has led to the decrease of their own athletic ability and physical quality. "inflexible", "unhealthy", etc. However, in real life, there is no obvious difference in the athletic ability of low fitness group and "weak" college students compared with normal students, and there is no big difference in the flexibility and reaction speed of these students compared with normal students, so the level of body self-perception is higher. Most Chinese college students think of body beauty as thinness, so the level of body perception is higher among low-fit college students and "weak" college students than obese and super-fit students. The experimental exercise intervention is aerobic exercise, which effectively burns fat and promotes weight loss, and effectively targets obese and super-fit college students, but not low-fit and "weak" college students, so on the contrary, perhaps strength and anaerobic training can improve low-fit and "weak" college students' body self-esteem. In contrast, perhaps strength and anaerobic training can improve the body self-esteem level of low and "weak" college students.

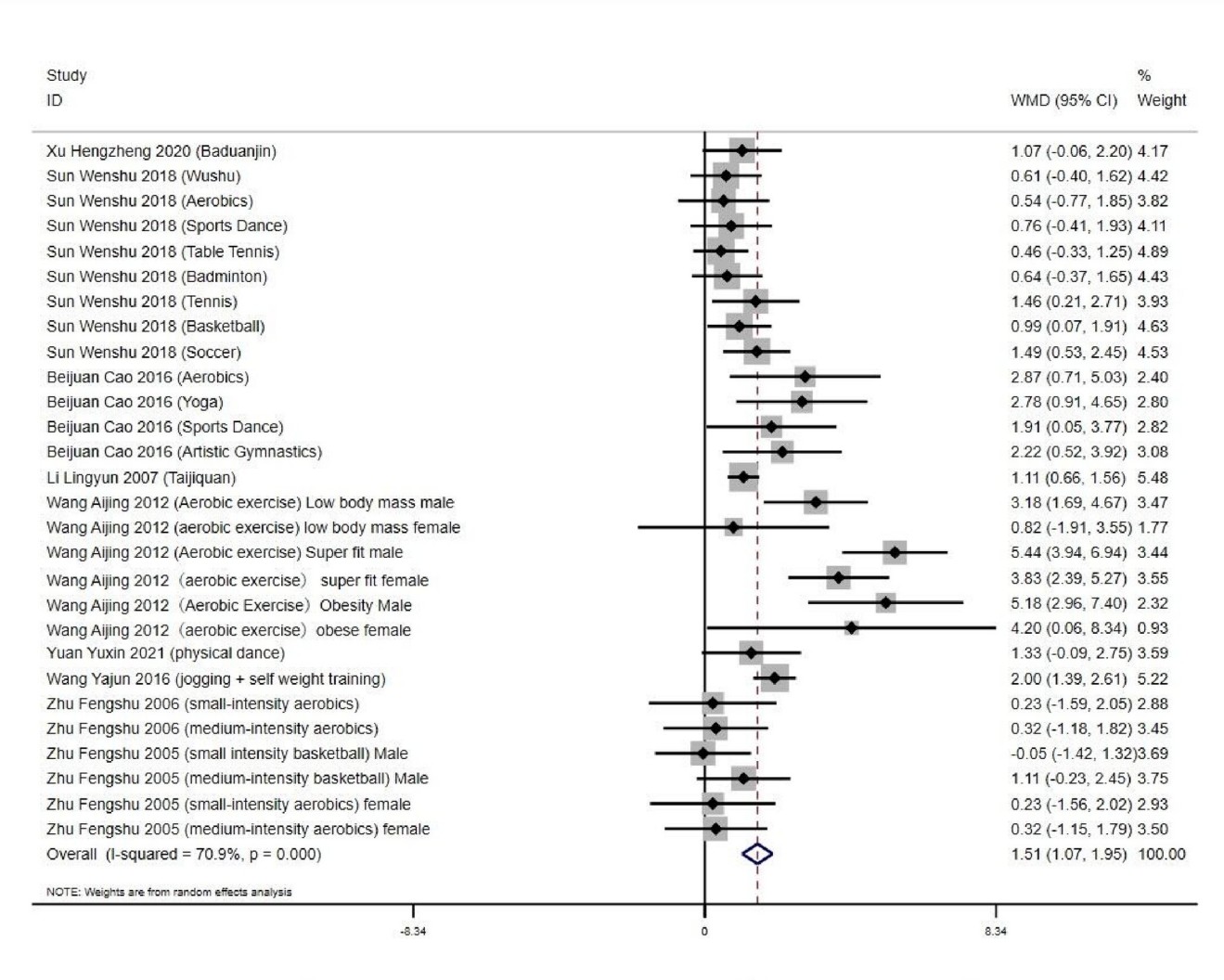

**Fig 11. Effect of aerobic exercise workout on physical fitness of college students.**

## 4.3 Effects of different aerobic exercise intensities on body self-esteem interventions for college students

Moderate intensity aerobic exercise promotes the level of body self-esteem of Chinese college students better than low intensity aerobic exercise, probably because moderate intensity aerobic exercise can better stimulate muscle growth and fat burning, build a beautiful body, improve athletic ability and physical fitness, and thus improve body self-esteem compared to low intensity aerobic exercise. However, the difference between moderate and low intensity aerobic exercise on body self-esteem intervention in some literature is not significant, and

**Table 9. Results of subgroup analysis of physical fitness.**

| Subgroup (Male/Female) | Number of studies | MD (95%CI) | P | Z | $I^2\%$ |
|---|---|---|---|---|---|
| Male | 8 | -0.64 [-0.96, -0.31] | P = 0.0001 | 3.87 | 81% |
| Female | 8 | -0.45 [-0.67, -0.23] | P<0.0001 | 4.00 | 48% |

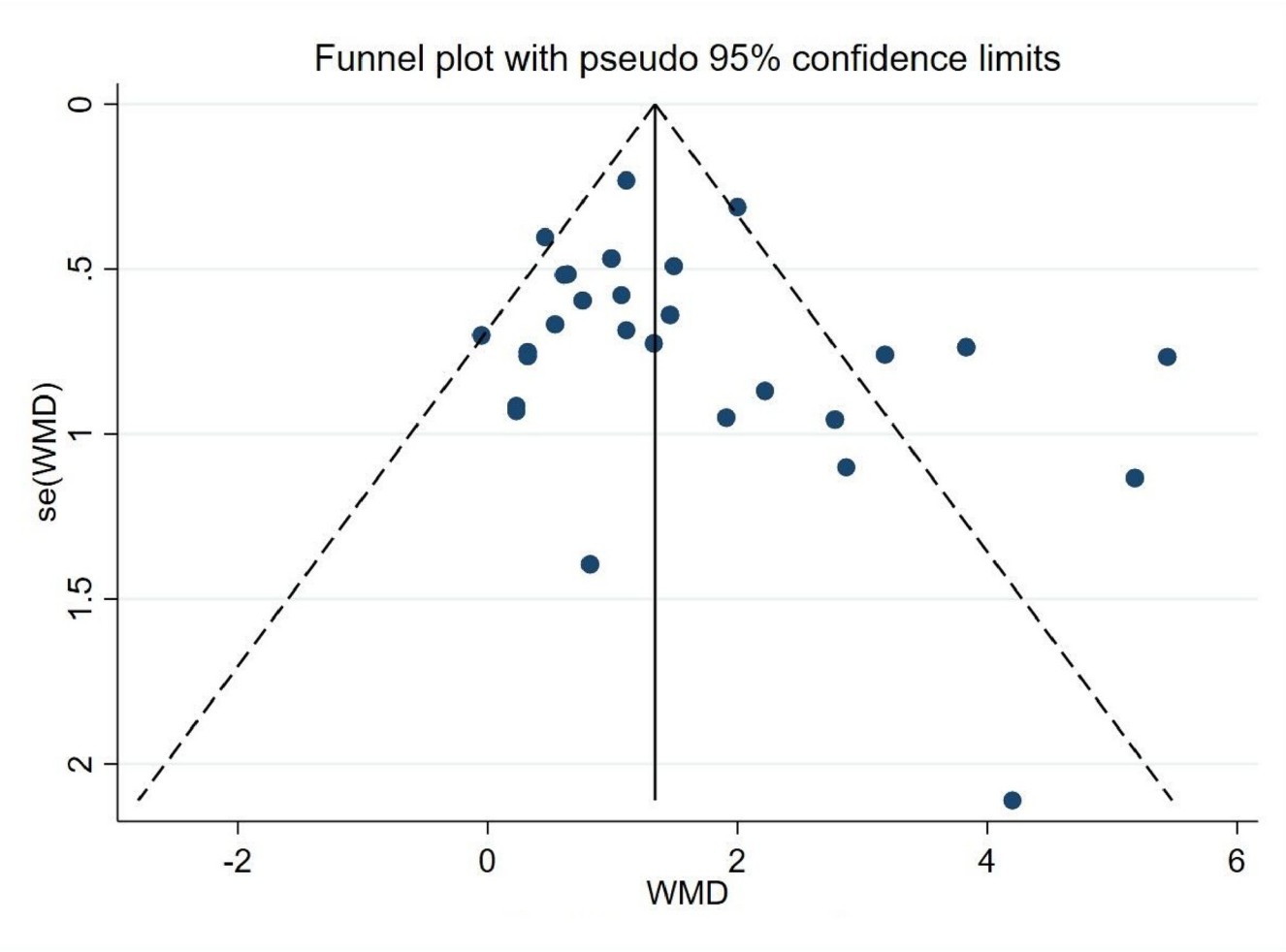

**Fig 12. Physical fitness funnel.**

even some data show that low intensity aerobic exercise is significantly more effective than moderate intensity aerobic exercise in intervening body self-worth and body condition, but the difference between the values of body self-worth and body condition after the two intensity interventions is less than 1, and the size of the difference is not significant, which may be because the difference between moderate intensity However, the difference between the two intensities was less than 1, and the size of the difference was not significant, which may be because the difference between medium intensity and small intensity was not obvious, and should be included in the comparison of large intensity exercise.

## 4.4 Effects of aerobic exercise of different types of sports on body self-esteem intervention of college students

All the exercise types included in the literature have a good promotion effect on Chinese college students' body self-esteem after the intervention, among which aerobics, physical dance, artistic gymnastics, and yoga have a good effect on college students' "athletic ability", "body self-worth", "physical condition", and "body self-esteem". The four dimensions of "physical condition" and "physical attractiveness" were improved more significantly. Perhaps because gymnastic exercises are accompanied by music, the movements are beautiful and rich, rich in

rhythm, so that the exerciser can transfer the tiredness brought by exercise, which invariably increases the consumption of exercise, improves the athletic ability and physical quality, while expressing their own emotions, improving aesthetic ability, and improving self-confidence when performing in front of others, gymnastic exercises are relatively more concerned with themselves, less competitive, and the cooperation, understanding, and help of the group. Collective cooperation, understanding, help and coordination are more important and can promote interpersonal relationships, so these four dimensions of improving body self-esteem are highly targeted and have a significant effect on improving.

In terms of improving "physical quality", basketball and soccer are more effective, probably because basketball and soccer require physical confrontation in sports, and they are more effective in improving physical quality because of the high exertion, long duration, and high stimulation of the body, while soccer and basketball also have better effects on the other 4 dimensions. Jogging plus self-weight training also has a better and very good effect on the 5 dimensions of body self-esteem, but not as good as gymnastic sports and basketball and soccer. Probably because jogging plus self weight training can also play a role in muscle building while effectively slimming down, the combination of the two consumes a lot, the practitioner focuses more on himself and improves the training effect. Taijiquan and Baduanjin are less effective in improving the various dimensions of body self-esteem than the previous exercises. These two exercises are low in intensity and have a slower and softer rhythm, which is less suitable for college students who are in their vigorous years, and therefore may affect the enthusiasm of the exercise, and the stimulation and energy consumption of the body is relatively small, so the effect of improving body self-esteem is limited.

### 4.5 Effects of aerobic exercise of different durations on body self-esteem intervention for college students

The latter has a better effect on improving body self-esteem compared with one workout per week of 90 minutes each, probably because it is aerobic exercise and each 30 minutes is not sufficiently stimulating, but each 90 minutes is a little more stimulating to the body and the exertion is increased. In terms of the overall duration of the intervention, one semester and one year of intervention are more effective than 16 and 10 weeks of intervention, but there may also be effects from different exercise programs and populations, and this needs to be demonstrated in later experiments.

### 4.6 Limitations of this study

(1) There is different publication bias and heterogeneity in the included data, probably due to the different duration of the intervention. (2) The collected literature spans a large range (3) There is no uniform requirement for the process at the time of measurement.(4)The included literature has limitations in the study population and lacks detailed information that may influence the findings, such as ethnic, geographical, and cultural factors.(5)Extraction of information is carried out independently by multiple researchers, but there may still be omissions due to subjective factors, affecting the comprehensiveness and authenticity of the evaluation results.(6) Fewer studies included in the literature, smaller sample sizes, and lack of stronger evidence for the results.

## 5. Conclusions

(1) Aerobic exercise can effectively improve the physical self-esteem of college students. (2) Male students pursue athletic ability and physical quality, while female students pursue a sense of physical self-worth and physical attractiveness. (3) The increase in exercise intensity also

improves body self-esteem. When aerobic exercise is performed, the intensity of exercise can be increased, but not the greater the intensity, but currently it can be proved that moderate intensity is good for body self-esteem intervention compared to small intensity exercise. (4) Aerobic exercise has a greater increase in body self-esteem for obese or super-fit college students. (5) Exercise class exercises are the most cost-effective for improving body self-esteem, and Taijiquan and Baduanjin are the least cost-effective. (6) An exercise session of 90 minutes was more effective than one session of 30 minutes in raising body self-esteem, and the overall intervention duration of 16 weeks was more effective than 10 weeks.

## Supporting information

**S1 Checklist. PRISMA 2020 checklist.**
(DOCX)

**S1 Data.**
(ZIP)

## Author Contributions

**Conceptualization:** Junwen Shu, Baole Tao, Jun Yan.

**Data curation:** Junwen Shu, Hanwen Chen.

**Formal analysis:** Haoran Sui, Lingzhi Wang.

**Funding acquisition:** Hanwen Chen.

**Investigation:** Tianci Lu, Hanwen Chen.

**Project administration:** Ye Zhang.

**Resources:** Baole Tao.

**Software:** Junwen Shu.

**Supervision:** Jun Yan.

**Validation:** Baole Tao.

**Visualization:** Junwen Shu.

**Writing – original draft:** Junwen Shu.

**Writing – review & editing:** Tianci Lu, Jun Yan.

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
