## [Decision Letter · Decision Letter 0]

11 Jul 2023

PONE-D-23-08546Effects of Aerobic Exercise on Body Self-Esteem among Chinese College Students:A Meta-AnalysisPLOS ONE

Dear Dr. Yan,

Thank you for submitting your manuscript to PLOS ONE. After careful consideration, we feel that it has merit but does not fully meet PLOS ONE’s publication criteria as it currently stands. Therefore, we invite you to submit a revised version of the manuscript that addresses the points raised during the review process.

We look forward to receiving your revised manuscript.

Kind regards,

Ru Zhang

Academic Editor

PLOS ONE

Journal Requirements:

Reviewers' comments:

Reviewer's Responses to Questions

**Comments to the Author**

1. Is the manuscript technically sound, and do the data support the conclusions?

Reviewer #1: Partly

2. Has the statistical analysis been performed appropriately and rigorously? 

Reviewer #1: No

3. Have the authors made all data underlying the findings in their manuscript fully available?

Reviewer #1: No

4. Is the manuscript presented in an intelligible fashion and written in standard English?

Reviewer #1: Yes

5. Review Comments to the Author

Reviewer #1: The study topic is worthy of investigation; however, this study has some problems that must be fixed before acceptance. Please find below some major points to consider.

1. Lines 51-52. You mentioned that psychologists consider body self-esteem the variable that best predicts changes in mood and personality. Please provide references to support it.

2. Line 99. You mentioned that your study followed the PRISMA statement. However, several sections in the checklist are missing, such as the selection process, synthesis method, results of synthesis, and registration. Please describe those missed sections in your manuscript and update the checklist.

3. Line 112. What kind of study design is eligible for your meta-analysis? Please describe the criteria in this section. Additionally, do you have any limitations on the language of included studies? Please describe it.

4. Lines 130-132. You mentioned using the pre-test data of the experimental group to replace the lacked data of the control group. Do you have any references to support this method?

5. If you followed the PRISMA statement. Please evaluate the certainty of each body of evidence, like using the GRADE approach.

6. I found that you include football and basketball in your meta-analysis. However, these two exercises cannot strictly be classified as aerobic exercises because they require both aerobic and anaerobic energy systems. Please consider excluding them from the analyses.

7. The heterogeneity (I square) of the results of all your meta-analysis is high. I am afraid that selection of the model (whether it is a fixed effect or a random effects model) has nothing to do with the statistical heterogeneity but with the underlying assumption. Please use appropriate methods to explain the high heterogeneity.

8. Lines 249-250. I did not find any subgroup analysis to compare different sexes in the results section. Please explain how did you get this conclusion.

6. PLOS authors have the option to publish the peer review history of their article (what does this mean?). If published, this will include your full peer review and any attached files.

Reviewer #1: No

---

## [Author Response · Author response to Decision Letter 0]

9 Aug 2023

Dear Editors and Reviewers:

Thank you for your letter and for the reviewers’ comments concerning our manuscript entitled “Effects of Aerobic Exercise on Body Self-Esteem among Chinese College Students: A Meta-Analysis”. Those comments are allvaluable and very helpful for revising and improving our paper, as well as theimportant guiding significance to our researches. We have studied comments carefullyand have made correction which we hope meet with approval. Revised portion aremarked in red in the paper. Some of your questions were answered below.

1.The reviewer’s comment: Lines 51-52. You mentioned that psychologists consider body self-esteem the variable that best predicts changes in mood and personality. Please provide references to support it.

The authors’ Answer: Thank you for your suggestion.With regard to lines 51-52, the meaning of the passage is "self-esteem" rather than " body self-esteem". Self-esteem is considered by psychologists to be the variable that best predicts changes in mood and personality. I have amended it and cited the relevant references.We apologise for this oversight!

2.The reviewer’s comment: Line 99. You mentioned that your study followed the PRISMA statement. However, several sections in the checklist are missing, such as the selection process, synthesis method, results of synthesis, and registration. Please describe those missed sections in your manuscript and update the checklist.

The authors’ Answer: Thank you for your suggestion.Errors in understanding the content of the PRISMA list at the time of writing led to the omission of several parts of the PRISMA list, which I have now re-added and updated.

3.The reviewer’s comment: Line 112. What kind of study design is eligible for your meta-analysis? Please describe the criteria in this section. Additionally, do you have any limitations on the language of included studies? Please describe it.

The authors’ Answer : Thank you for your suggestion.The included studies are experimental exercise intervention studies in English or Chinese, but the subjects must be Chinese university students. Revised "2.2 Literature inclusion criteria".

4.The reviewer’s comment: Lines 130-132. You mentioned using the pre-test data of the experimental group to replace the lacked data of the control group. Do you have any references to support this method?

The authors’ Answer : Thank you for your suggestion.We queried for studies with approaches to dealing with missing data from Meta-analyses, and a study published in Research Synthesis Methods (IF 9.8) illustrates the feasibility of this approach. Similar studies using this approach were also found. References have been included in the article.

5.The reviewer’s comment: If you followed the PRISMA statement. Please evaluate the certainty of each body of evidence, like using the GRADE approach.

The authors’ Answer : Thank you for your suggestion.In accordance with your request, the results of the GRADE system grading are now presented in Table 4, in rows 188 - 191, which I think is necessary.

6.The reviewer’s comment: I found that you include football and basketball in your meta-analysis. However, these two exercises cannot strictly be classified as aerobic exercises because they require both aerobic and anaerobic energy systems. Please consider excluding them from the analyses.

The authors’ Answer : Thank you for your suggestion.Aerobic exercise is exercise that is low in intensity but long and rhythmic. Its main characteristics are low intensity, rhythmic, uninterrupted and long duration, while keeping the heart rate below 150 beats per minute. Both football and basketball were selected for our study where heart rate was controlled and both lasted longer than 30min, so they can be classified as aerobic exercise. However this suggestion made us realise that we did not express the number of heart rates per minute in the paper. This is now presented in "Table 2 Summary of studies that met the inclusion criteria".

7.The reviewer’s comment: The heterogeneity (I square) of the results of all your meta-analysis is high. I am afraid that selection of the model (whether it is a fixed effect or a random effects model) has nothing to do with the statistical heterogeneity but with the underlying assumption. Please use appropriate methods to explain the high heterogeneity.

The authors’ Answer : Thank you for your suggestion.In my opinion, there are the following problems with heterogeneity: 1. Different genders have different pursuit of the five dimensions of body self-esteem. 2. The duration of various aerobic exercise interventions is different, and the frequency of intervention is different. 3. Multiple types of aerobic exercise, resulting in results that are not consistent.

8.The reviewer’s comment: Lines 249-250. I did not find any subgroup analysis to compare different sexes in the results section. Please explain how did you get this conclusion.

The authors’ Answer : Thank you for your suggestion.I came to this conclusion because I divided the various studies into groups, where these groups included different genders, and I came to this conclusion by comparing the different genders. However, I decided to take the advice you gave me because I thought it was not rigorous enough and the advice you gave was better. I have added a subgroup analysis of gender to my analysis of each result. Presented in tables 4 to 8.

---

## [Decision Letter · Decision Letter 1]

22 Aug 2023

Effects of Aerobic Exercise on Body Self-Esteem among Chinese College Students: A Meta-Analysis

PONE-D-23-08546R1

Dear Dr. Yan,

We’re pleased to inform you that your manuscript has been judged scientifically suitable for publication and will be formally accepted for publication once it meets all outstanding technical requirements.

Kind regards,

Ru Zhang

Academic Editor

PLOS ONE

Additional Editor Comments (optional):

Reviewers' comments:

Reviewer's Responses to Questions

**Comments to the Author**

1. If the authors have adequately addressed your comments raised in a previous round of review and you feel that this manuscript is now acceptable for publication, you may indicate that here to bypass the “Comments to the Author” section, enter your conflict of interest statement in the “Confidential to Editor” section, and submit your "Accept" recommendation.

Reviewer #1: All comments have been addressed

2. Is the manuscript technically sound, and do the data support the conclusions?

Reviewer #1: Yes

3. Has the statistical analysis been performed appropriately and rigorously? 

Reviewer #1: Yes

4. Have the authors made all data underlying the findings in their manuscript fully available?

Reviewer #1: Yes

5. Is the manuscript presented in an intelligible fashion and written in standard English?

Reviewer #1: Yes

6. Review Comments to the Author

Reviewer #1: The authors have addressed all the comments raised by me. I'm satisfied with the response. The manuscript now is now ready for publication.

7. PLOS authors have the option to publish the peer review history of their article (what does this mean?). If published, this will include your full peer review and any attached files.

Reviewer #1: No

---

## [Editor Report · Acceptance letter]

29 Aug 2023

PONE-D-23-08546R1 

Effects of Aerobic Exercise on Body Self-Esteem among Chinese College Students: A Meta-Analysis 

Dear Dr. Yan:

I'm pleased to inform you that your manuscript has been deemed suitable for publication in PLOS ONE. Congratulations! Your manuscript is now with our production department. 

Kind regards, 

on behalf of

Dr. Ru Zhang 

Academic Editor

PLOS ONE